# Real-Time Longitudinal Evaluation of Tumor Blood Vessels Using a Compact Preclinical Fluorescence Imaging System

**DOI:** 10.3390/bios11120471

**Published:** 2021-11-23

**Authors:** Hoibin Jeong, Song-Rae Kim, Yujung Kang, Huisu Kim, Seo-Young Kim, Su-Hyeon Cho, Kil-Nam Kim

**Affiliations:** 1Chuncheon Center, Korea Basic Science Institute (KBSI), Chuncheon 24341, Korea; hbj04@kbsi.re.kr (H.J.); ksr87@kbsi.re.kr (S.-R.K.); kimsy11@kbsi.re.kr (S.-Y.K.); chosh93@kbsi.re.kr (S.-H.C.); 2Vieworks, Anyang 14055, Korea; yujung.kang@vieworks.com (Y.K.); hskim@vieworks.com (H.K.); 3Division of Practical Application, Honam National Institute of Biological Resources, Mokpo 58762, Korea; 4Department of Bio-Analytical Science, University of Science and Technology, Daejeon 34113, Korea

**Keywords:** optical imaging, indocyanine green, preclinical study, tumor angiogenesis, blood flow

## Abstract

Tumor angiogenesis is enhanced in all types of tumors to supply oxygen and nutrients for their growth and metastasis. With the development of anti-angiogenic drugs, the importance of technology that closely monitors tumor angiogenesis has also been emerging. However, to date, the technology for observing blood vessels requires specialized skills with expensive equipment, thereby limiting its applicability only to the laboratory setting. Here, we used a preclinical optical imaging system for small animals and, for the first time, observed, in real time, the entire process of blood vessel development in tumor-bearing mice injected with indocyanine green. Time-lapse sequential imaging revealed blood vessel volume and blood flow dynamics on a microscopic scale. Upon analyzing fluorescence dynamics at each stage of tumor progression, vessel volume and blood flow were found to increase as the tumor developed. Conversely, these vascular parameters decreased when the mice were treated with angiogenesis inhibitors, which suggests that the effects of drugs targeting angiogenesis can be rapidly and easily screened. The results of this study may help evaluate the efficacy of angiogenesis-targeting drugs by facilitating the observation of tumor blood vessels easily in a laboratory unit without large and complex equipment.

## 1. Introduction

Angiogenesis progresses in solid tumors to provide the oxygen and nutrients necessary for the continuous growth of the tissue and metastasis [1]. Therefore, research on drugs targeting angiogenesis as an effective strategy for cancer treatment is ongoing. Several angiogenesis inhibitors are undergoing clinical trials, and some are already being used in the clinical setting [2]. However, the effects of these agents differ from tumor types or patient to patient, and are still limited because of the tumor resistance mechanisms [3]. In response to angiogenesis inhibitors, vascular elements such as vessel volume, blood flow, and microvascular density change simultaneously and dynamically, and these changes occur over several days [4]. Therefore, their local changes have the potential to be biomarkers in cancer treatment, and a method to measure them longitudinally is essential for judging therapeutic effects. To screen drugs against angiogenesis and to predict their therapeutic effects, it is important to develop a practical tool that can repeatedly observe and measure vascular changes in tumors.

Several other studies have shown detailed real-time vasculature progression based on fluorescence imaging. Blood vessel dysmorphia characterized by dramatic changes in vessel expansion, leakage, and loss of branching complexity during glioma growth was elucidated using high-resolution two-photon microscopy [5]. Alterations in tumor vasculatures were observed for improving drug delivery and antitumor responses in a melanoma mouse model using intravital microscopy [6]. Bochner’s group developed a novel method for longitudinal intravital imaging of vascular and tumor microenvironment remodeling in spontaneous and orthotopic pancreatic tumor models [7]. Although these imaging methods using fluorescent dyes enable researchers to observe and analyze blood vessels in tissues such as brain or tumor tissues, there is a limitation in that they require the insertion of a window chamber into a target organ by surgical operations, which is not applicable to the patients that we aim to study. Other techniques to monitor the shape of blood vessels and its formation process are laser Doppler velocimetry and magnetic resonance imaging (MRI), which do not involve surgical procedures [8,9]. However, their use is limited because of the requirement of a bulky and costly hardware system equipped with sophisticated analysis software, making them unsuitable for installation and use in a laboratory unit. According to a recent study, the direction and quantitative velocity of blood flow in vivo are measurable by combining particle image velocimetry with a speckle-based method [10]. Despite the advantage of not requiring the injection of an exogenous reagent and a surgical operation for imaging, compared to fluorescence imaging, this technique has limitations in that spatial resolution and imaging depth are inferior [11,12,13], and it is difficult to determine various vascular parameters at once in a non-invasive strategy. Hence, the introduction of techniques that rapidly and easily identify various vascular elements would substantially help in evaluating the efficacy of angiogenesis inhibitors.

For that purpose, indocyanine green (ICG), a fluorescent dye excited in the near-infrared (NIR) wavelength, is often implemented. Because of its complete excretion by the liver and its high safety, ICG is used in the clinical examination of the liver function of patients before surgery [14]. Recently, ICG was utilized to visualize the structure of target organs and to identify tumors in vivo with the enhanced permeability and retention effect of ICG, a phenomenon in which nanoscale molecules selectively penetrate abnormal capillaries of cancer tissues and, at the same time, accumulate in cancer tissues because of ineffective excretion into degenerated lymphatic vessels. Combining pharmacokinetics of ICG in various animal models with principal component analysis allows researchers to successfully discriminate various organs [15], distinguish tumors from normal tissues [16], and track pulmonary vascular permeability in the lungs caused by radiation injury [17]. Jagtap et al. reported that genetic modifiers affecting vascular function could be identified by injecting ICG into transgenic rats transplanted with breast cancer [18]. Choi’s group observed the vascular density, perfusion rate, and permeability of tumors and compared them to those in normal tissues using dynamic fluorescence imaging with ICG [19]. Since they analyzed vascular parameters only at a specific tumor growth stage, however, dynamic changes in blood vessels throughout the entire process of tumor growth could not be observed.

In this study, we present a novel technique to observe even the microscopic blood vessels in mouse subcutaneous tumors using a compact preclinical fluorescence imaging system called VISQUE^®^ InVivo Smart-LF, developed by Vieworks Co, Ltd. (Anyang-si, Korea). Of the optical in vivo imaging systems for small animals, only Smart-LF possesses a time-lapse imaging function and, at the same time, superior spatial resolution (28 μm at 3-fold magnification) and imaging depth (Appendix A). We monitored the changes in vascular development and analyzed the blood volume and flow in the vessels at each stage of tumor development by injecting ICG into mice. By applying the high-speed time-lapse imaging function, it was possible to observe the blood flow in real time as a video. In addition, we evaluated the effects of an angiogenesis inhibitor on the parameters mentioned above, suggesting that this technique could be employed to quickly and easily screen candidate angiogenesis-targeting agents.

## 2. Materials and Methods

### 2.1. Animal Preparation

Five-week-old BALB/c-nude female mice (Nara-Biotec, Seoul, Korea) were maintained under controlled conditions of temperature (23 ± 2 °C), humidity (55 ± 5%), and light (12 h light/dark cycle) at the Korea Basic Science Institute (KBSI), and had access to food and water *ad libitum*. All animal experiments were approved by the Institutional Animal Care and Use Committee at the KBSI.

### 2.2. Cell Culture

A431 (a human epidermoid carcinoma cell line) was cultured in Roswell Park Memorial Institute medium (HyClone Laboratories Inc., Logan, UT, USA) supplemented with 10% fetal bovine serum (Omega Scientific Inc., Tarzana, CA, USA), penicillin (100 U/mL), and streptomycin (100 μg/mL; Invitrogen, Carlsbad, CA, USA), and maintained at 37 °C in a 5% CO_2_ incubator.

### 2.3. Tumor Growth and Therapy Dosage

For the mouse xenograft model, 1 × 10^6^ cells of A431 were subcutaneously implanted, and tumor size was calculated using a formula for hemi-ellipsoid (volume = 0.5236 × length × width × height) using calipers. According to the size of the tumor, it was divided into five grades (Grade I, size < 20 mm^3^ and palpable; Grade II, size ~50 mm^3^; Grade III, size ~120 mm^3^; Grade IV, size ~300 mm^3^; Grade V, size > 500 mm^3^). As a control group for tumor-bearing mice, a Matrigel (Corning, NY, USA) with a size similar to that of the tumor was implanted subcutaneously at the same location as the tumor. For the vascular endothelial growth factor (VEGF) inhibition experiment, 4 μg of mVEGFR1 Ab or IgG control (R&D Systems, Minneapolis, MN, USA) was injected intratumorally twice every three days when the tumor size reached 120 mm^3^.

### 2.4. ICG Tumor Imaging

When the tumor size reached the designated grade, ICG (2.5 μg/g; Dongindang Pharmaceutical, Siheung-si, Korea) was injected into the respiratory anesthetized mouse intravenously using a catheter with a 29-gauge needle (Instech Laboratories, Plymouth Meeting, PA, USA). For intravenous catheterization, the needle was inserted into the tail vein of the mouse and fixed with tape. A syringe barrel without a needle (BD, Franklin Lakes, NJ, USA) containing ICG was attached to a syringe pump and connected with the catheter (Figure 1). The syringe pump (Braintree Scientific Inc., Braintree, MA, USA) was used to inject a certain amount of ICG at a constant rate (2.5 mL/min). At the same time as ICG was injected, the tumor was illuminated by 760 nm wavelength light. The fluorescence dyes with the same wavelength as ICG have a penetration depth of about 5 mm in the tissue [20], which is long enough for the subcutaneous tumor to come into focus at once. Time-sequence images of ICG fluorescence signals were acquired under the conditions of 3-fold magnification conditions (field-of-view of 5 cm × 5 cm), axial resolution of 27 μm, and longitudinal resolution of about 6.7 frames per second (150 ms intervals) for 75 sec under 100 ms exposure time using VISQUE^®^ InVivo Smart-LF (Vieworks. Co, Ltd., Anyang-si, Korea), an optical in vivo imaging system for small animals with fluorescence and luminescence imaging capabilities. This device is equipped with an sCMOS camera with an 830 nm wavelength filter, providing an imaging speed of up to 37 frames per second with an exposure time of 25 ms to 15 min. The light source used was light emitting diode. Specifications for sCMOS camera, ‘VSP-4MU’ manufactured by Vieworks Co, Ltd., and VISQUE^®^ InVivo Smart-LF equipment are detailed in Appendix A.

### 2.5. Analysis of Fluorescence Dynamics

In the fluorescence images, the blood vessels could be visualized as bright areas under the proper excitation by light and emission filter after ICG was injected into a blood vessel. Blood flow maps were analyzed and generated from the time-lapse images using the software provided by the manufacturer (Vieworks Co, Ltd.). The algorithm for blood flow analysis followed the methods of Ku and Kalchenko [21,22]. In short, blood flow dynamics were measured for maximum fluorescence intensity (*I*_max_), *I*_arrival_, *T*_rising_, and blood flow index (BFI) at each pixel in time-lapse images. As shown in Figure 2, *I*_arrival_ means the fluorescence intensity value when the fluorescence intensity above the threshold value first appears in each pixel. *T*_rising_ is calculated as the duration between *T*_arrival_ (the time when ICG fluorescence first appears) and *T*_max_ (the time when maximum intensity is). The BFI was calculated by dividing the relative fluorescence intensity between *I*_max_ and *I*_arrival_ by *T*_rising_, with the index representing overall blood volume information over time. FluAngio was made by the calculation of change of fluorescence intensity. During time-lapse imaging, the surface can move due to breathing, and the fluorescence intensity of the surface also changes, resulting in a white contrast on the surface. For analyzing various parameters of tumor blood vessels by grade, the thickest part of the vasculature that appears to be the same for each grade is selected, and then the region of interest (ROI) is drawn locally so that only the selected vascular region is included (Appendix A). To correct the movement by the respiratory motion of the animal, breathing artifacts were removed from the signal of each pixel using Savitzky–Golay Filter (Appendix A).

### 2.6. Immunofluorescence Staining

Tumors were harvested and prepared as frozen sections. Sections on slides were fixed using 4% paraformaldehyde (DaeJung Chemicals, Siheung-si, Korea) for 30 min. For detecting blood vessels in tumor tissues, cluster of differentiation 31 (CD31), a marker of vascular endothelial cells, was used as a target molecule. Sections on fixed slides were incubated with anti-CD31 (rabbit anti-human CD31 monoclonal antibodies; Thermo Fisher Scientific, Waltham, MA, USA) for 2 h at 24 °C. Secondary antibodies were anti-rabbit Alexa 488 (Life Technologies, Carlsbad, CA, USA), which were incubated for 45 min at 24 °C. After washing with phosphate-buffered saline, sections were counterstained with Hoechst 33,342 (Sigma-Aldrich, Saint Louis, MO, USA) at 10 μg/mL for 10 min at 24 °C and examined using LSM 780 Zeiss Confocal Laser Microscope (Zeiss, Oberkochen, Germany). Digital images were processed using the ZEN 2010 software and quantified by directly counting the cells stained with CD31 on the cell surface.

### 2.7. Statistical Analysis

Data were statistically compared using two-tailed Student’s *t*-test for two independent groups, two-tailed one-way ANOVA with Tukey’s post-test for three or more independent groups with only one independent factor, or two-way ANOVA with Bonferroni correction for three or more independent groups with two independent factors using Prism software (Version 8; GraphPad Inc., San Diego, CA, USA). Data were considered statistically significant at *p* < 0.05.

## 3. Results

### 3.1. Time-Lapse Sequential Imaging and Kinetic Analysis of Blood Vessels in Tumors Using VISQUE^®^ In Vivo Smart-LF

To observe the tumor vessels in real time, time-lapse sequential imaging was acquired while ICG was injected into mice implanted with A431 human skin squamous carcinoma cells. The results of time-lapse images confirmed that blood vessels inside the tumor were clearly visible (Figure 3a and Appendix A). In contrast, in the Matrigel implanted subcutaneously to simulate the shape of a tumor, no blood vessels were observed. The dynamics of fluorescence intensity in the entire area of the tumor or Matrigel over time increased rapidly at the beginning and then persisted (Figure 3b).

*I*_max_ and the BFI were calculated from these fluorescence dynamics graphs for quantitatively assessing the blood flow dynamics. Both *I*_max_ and the BFI were substantially higher in tumors compared to Matrigels (Figure 3c). In addition, microscopic blood vessels with a thickness of 0.2 mm in the tumor were also observed through the FluAngio map, which represents the path blood vessels pass through (Figure 3d). We further confirmed that the blood vessels within the tumor or tumor periphery (skin) regions clearly visible with the naked eye in the photographs were the same as those seen in *I*_max_ and FluAngio obtained from ICG imaging (Appendix A). This means that blood vessels can be easily observed and analyzed kinetically in vivo without any invasion in real time using ICG and a compact device.

### 3.2. Monitoring Tumor Angiogenesis and Analyzing Vascular Parameters throughout the Entire Tumor Growth Process

To monitor the entire process of angiogenesis, we divided tumor growth into five grades according to the tumor size, and time-lapse imaging for each grade was performed. No blood vessels were observed in Grade I; however, they appeared in Grade II, and the shape of the blood vessels dynamically changed as the tumor grew (Figure 4a and Appendix A). The entire tumor regions were designated as the ROI and analyzed to measure the blood flow changes according to the growth of the tumor. The overall fluorescence intensity of the tumor increased as the tumor grew; however, the changes could not be properly quantified though they were clearly visible (Appendix A). Conversely, when the ROI was localized to the thickest region where the accumulation of blood vessels begins in the tumor (Appendix A), *I*_max_ and the BFI, calculated from the dynamics of each ROI (Figure 4b), substantially increased as the tumor progressed (Figure 4c). A FluAngio map confirmed the increased size of the tumor and the number of primary or micro-blood vessels inside the tumor (Figure 4d).

Immunostaining in tumor areas without prominent blood vessels showed that the number of CD31-positive vascular endothelial cells decreased as the tumor progressed (Figure 4e), consistent with the exacerbation of hypoxia during tumor progression [23]. Opposite changes in fluorescence intensity were seen in the localized area without prominent blood vessels depending on the grade (Figure 4b,f). *I*_max_ and the BFI also rapidly decreased as the grade increased (Figure 4g). This means that as the tumor grows, the thickness of blood vessels and blood flow in the main vessel area increase, whereas they decrease in the area with tiny vessels, which is non-detectable with the ICG technique.

### 3.3. Evaluation of the Effect of VEGF Inhibition on Vascular Parameters

To examine whether the effect of VEGF inhibition can be analyzed by changes in blood flow dynamics, mVEGFR1 Ab was injected intratumorally when the tumor reached the Grade III stage (approximately 120 mm^3^). mVEGFR1 Ab treatment twice every three days reduced the growth of the tumor (Appendix A) and substantially decreased the number of CD31-positive cells (Figure 5a). To compare the kinetics of blood flow between the IgG and mVEGFR1 Ab groups, we designated the ROI as the main vessel part (Figure 5b). The graph showing the intensity of the ROI at each time point showed a sharp increase in the intensity over time in the IgG group, whereas such an increase was not detected in the mVEGFR1 Ab group (Figure 5c). *I*_max_ and the BFI also increased over time in the IgG, as shown in previous data from the entire tumor growth process (Figure 4c); however, *I*_max_ was decreased and the BFI remained unchanged in the mVEGFR1 Ab group (Figure 5d). These results demonstrated that the inhibition of angiogenesis by mVEGFR1 Ab decreased the blood vessel volume and blood flow, which could be observed using our imaging strategy; thus, the effect of angiogenesis inhibitors can be longitudinally screened in real time in vivo.

## 4. Discussion

In most solid tumors, angiogenesis is an essential process for their growth and metastasis. Therefore, studies on the mechanism underlying the process of angiogenesis and the development of drugs inhibiting it have been actively conducted. In this study, we non-invasively and longitudinally monitored the entire process of angiogenesis in the subcutaneous tumor in a xenograft mouse model using a compact optical imaging device. In addition, by observing the changes in vascular parameters after the administration of angiogenesis inhibitors, we validated the potential of this imaging technique in determining the efficacy of drugs targeting tumor vasculature.

Tumor vasculature could be simply analyzed by immunostaining in surgically harvested tissues; however, real-time longitudinal study of tumors has been challenging. Several imaging systems have been developed for the time-lapse assessment of tumors. Implantation of a surgical window chamber where tumors are generated allows us to observe vascular changes in vivo with high resolution, [24], although it is invasive and terminal. As a non-invasive imaging method, the ultrasound technique can evaluate blood vessels in tumors without surgery and can be applied in clinical trials [25]. However, visualization of microvascular changes by ultrasound is challenging without microbubble contrast agents. Similarly, MRI requires the use of contrast agents for high-resolution images [9], restricting longitudinal studies due to the effects of the magnetic field on metal devices implanted in a living body.

Optical imaging techniques for estimating tumor blood vessels enable time-series measurement, increasing attention on them as an alternative. Laser Doppler detects blood flow by measuring the circulation of red blood cells [26]; however, it is not sufficiently sensitive for high-resolution results. ICG, a NIR fluorescence probe widely used to measure in vivo vascular imaging, has a short half-life, low auto-fluorescence, and high tissue penetration. With these advantages, it has been used to perform measurement of tumor vasculature elements such as perfusion, permeability, and vessel density in mouse or rat tumors [18,19] and to observe blood flow in organs including the brain, hindlimb, and lungs [17,27,28]. Although optical imaging with ICG enables non-invasive measurement of tumor vasculature, there has been no study that has observed even microscopic blood vessels while performing multiparametric measurement. Here, we utilized compact optical imaging equipment and convenient analysis software to observe micro-blood vessels (0.2 mm) in real time with high resolution, and evaluated vascular changes throughout the entire process of tumor growth.

In this research, we divided tumor growth into five grades to assess the entire process of tumor growth. However, in this process, as the tumor grows, the ROI area also increases, making it difficult to quantify the visible changes. In addition, the shape of the tumor vessels varies for every individual, which makes the designation of an ROI location challenging. This limitation was not a concern in studies that observed cerebrovascular vessels in the stroke model [27] or the femoral artery in the hindlimb ischemia model [28], which used the same experimental techniques because the ROI region was not reported to change when observed longitudinally. We confirmed that, for the same subject, even if the tumor size continued to change, it was safe to designate localized blood vessel regions and follow-up (Appendix A).

We found that the level of blood flow and the thickness of the main vessels increased as the tumor grew. The prominent major blood vessels began to be visible from the Grade II tumor. After the progression to Grade IV, the levels of blood flow and the thickness of vessels did not increase further. It has been reported that as the tumor grows, vessel remodeling occurs, making them highly permeable, and small microvessels branch rapidly, causing chaotic and slow blood flow [29]. We found that the number of microvessels analyzed in FluAngio maps continuously increased depending on the grade, and the most severe increase was observed in Grade V, which explains why the trend of increasing ICG intensity continued to be steep up to Grade IV, but why in Grade V the intensity appeared to be moderate, as seen in Figure 4c. In the area with few blood vessels, *I*_max_ and blood flow gradually decreased, consistent with the immunofluorescence staining, confirming that the number of CD31-positive vascular endothelial cells decreases by grade. This means that as the tumor progresses, the size of the main vessels and the amount of blood entering the tumor tissues increase, whereas the blood supply is restricted in the areas which few microvessels reach, and hypoxia can be observed [30].

Tumors create new blood vessels through angiogenesis to obtain oxygen and nutrients for their growth. VEGF is one of the most important cytokines that induces angiogenesis in tumors [31], and VEGF inhibitors alter the dynamics of tumor vessels. We found that when angiogenesis was suppressed by mVEGFR1 Ab, the thickness and blood flow in the main blood vessels of the tumor decreased compared to the IgG treatment. This is consistent with the findings of MRI observation in a previous study, namely that the blood volume and vessel density decreased in mouse subcutaneous tumors when they were treated with anti-VEGF [32].

## 5. Conclusions

In summary, we developed a novel imaging system to observe the entire process of tumor angiogenesis and to analyze the vascular dynamics in real time. As the tumor progressed, the dynamics of the blood vessel shape were captured, and the vessel volume and blood flow at each time point were measured by analyzing the dynamics. This was performed using a relatively compact fluorescence imaging system, which is more convenient to use than existing angiography equipment. In addition, changes in vascular features could be evaluated after the administration of angiogenesis inhibitors, indicating that our technique can be utilized in high-throughput screening of drugs that target tumor angiogenesis.

## Figures and Tables

**Figure 1 biosensors-11-00471-f001:**
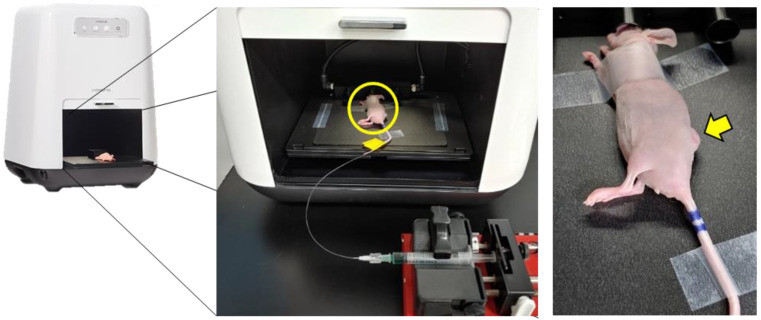
Schematic diagram of the ICG injection for time-lapse imaging in A431 tumor-bearing mice. The yellow circle in the middle panel indicates a mouse whose tail is fixed with a catheter attached to the syringe pump. The yellow arrow indicates a subcutaneous tumor to be imaged.

**Figure 2 biosensors-11-00471-f002:**
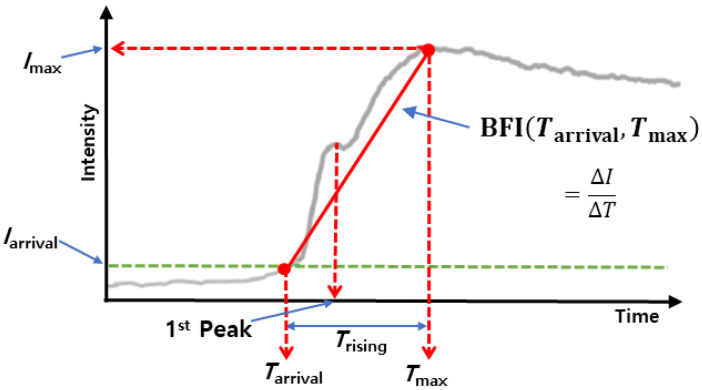
ICG dynamics with vascular parameters displayed. The red solid line indicates the slope of the peak time, BFI. *I*_arrival_, the threshold value, is determined by one-tenth of *I*_max_. The BFI was calculated by dividing the relative fluorescence intensity between *I*_max_ and *I*_arrival_ by *T*_rising_.

**Figure 3 biosensors-11-00471-f003:**
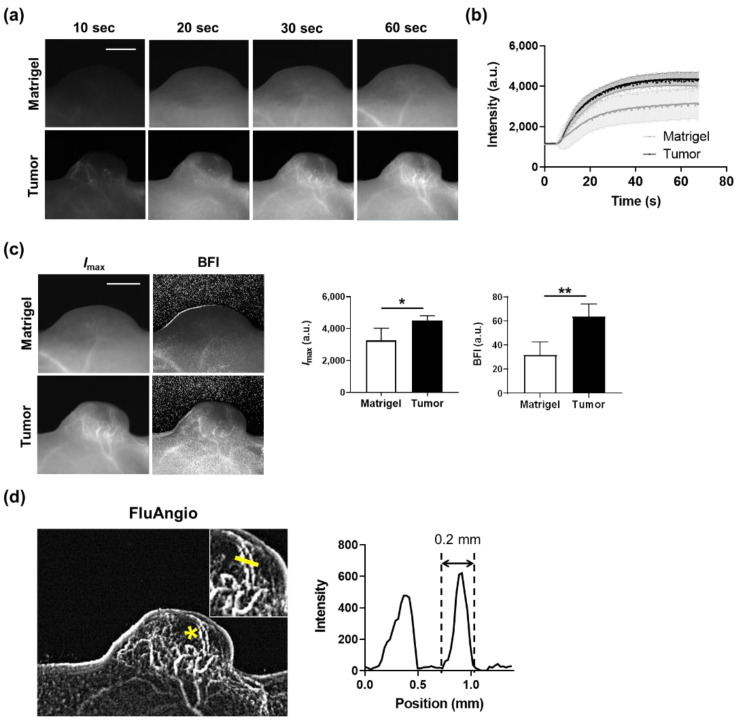
Real-time imaging of tumor vasculature using ICG and kinetic analysis. (**a**) Time-series acquisition of fluorescence images in Matrigel (top) and tumor (bottom) after intravenous injection of ICG. Scale bar, 5 mm. (**b**) ICG dynamics in the whole area of Matrigel or tumor. Data are the mean ± SD for *n* = 4 mice per group. (**c**) Representative maps were drawn using the *I*_max_ or BFI parameter in mice subcutaneously implanted with Matrigel or tumor (**left**). Scale bar, 5 mm. Quantification of *I*_max_ and BFI in the whole area of Matrigel and tumor (**right**). Data are the mean ± SD for *n* = 4 mice per group. *, *p* < 0.05; **, *p* < 0.01 by Student’s *t*-test. (**d**) A FluAngio map clearly showing the shape of blood vessels in a tumor (left). The inset shows the magnified region indicated with an asterisk (*). Line profile graph representing the fluorescence intensity above the yellow line crossing microvessels inside the inset shows that it can measure blood microvessels as small as 0.2 mm (right).

**Figure 4 biosensors-11-00471-f004:**
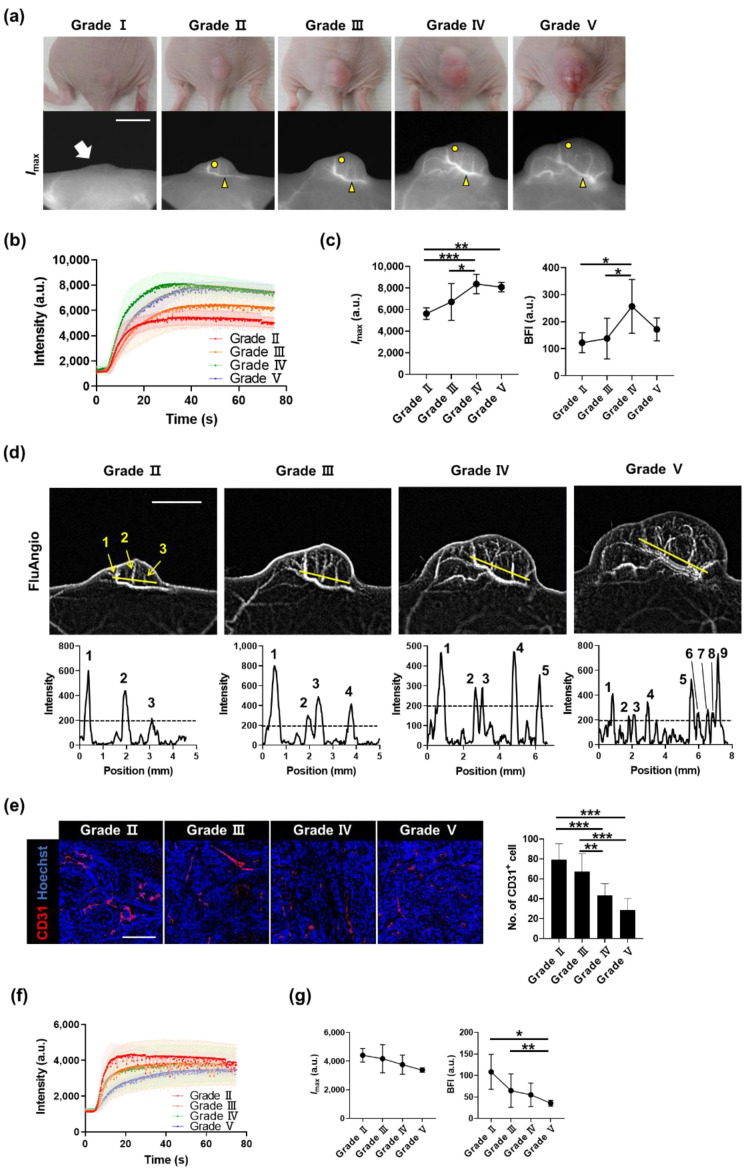
Monitoring the whole process of tumor angiogenesis and its kinetic analysis. (**a**) A431 tumors and fluorescence images for each grade were determined based on the tumor size. The white arrow on Grade I indicates the location of the small palpable tumor. In Grades II, III, IV, and V, yellow arrowheads indicate blood vessels analyzed in (**b**,**c**), and yellow circles indicate outer areas analyzed in (**f**,**g**). Scale bar, 5 mm. (**b**) ICG dynamics of blood vessels on tumors, marked by yellow arrowheads in (**a**), from each grade. (**c**) Changes in *I*_max_ and BFI in blood vessels by grade. Data are the mean ± SD for *n* = 8 mice per grade. *, *p* < 0.05; **, *p* < 0.01; ***, *p* < 0.001 by one-way ANOVA. (**d**) FluAngio maps presenting tumor vessels on each grade (top). As the size of the tumor increases, the length of the line (yellow) is proportionally increased. It can be seen that the three micro blood vessels clearly visible on the FluAngio map of Grade II can also be identified in the line profile. Scale bar, 5 mm. Line profile graphs of fluorescence intensity above yellow lines on FluAngio maps show the changes in tumor microvessels and an increase in their number as the grade progressed (bottom). (**e**) Immunostaining of CD31 (red) in outer regions from A431 tumors, marked by yellow circles in (**a**), on each grade (left), and the number of CD31-positive cells (right). Nuclei are stained in blue with Hoechst 33,342 counterstaining. Data are the mean ± SD for at least two independent fields examined per mouse (*n* ≥ 3 mice per group). Scale bar, 100 μm. **, *p* < 0.01; ***, *p* < 0.001 by one-way ANOVA. (**f**) ICG dynamics of outer areas on tumors from each grade. (**g**) Changes in *I*_max_ and BFI in outer areas by grade. Data are the mean ± SD for *n* = 8 mice per grade. *, *p* < 0.05; **, *p* < 0.01 by one-way ANOVA.

**Figure 5 biosensors-11-00471-f005:**
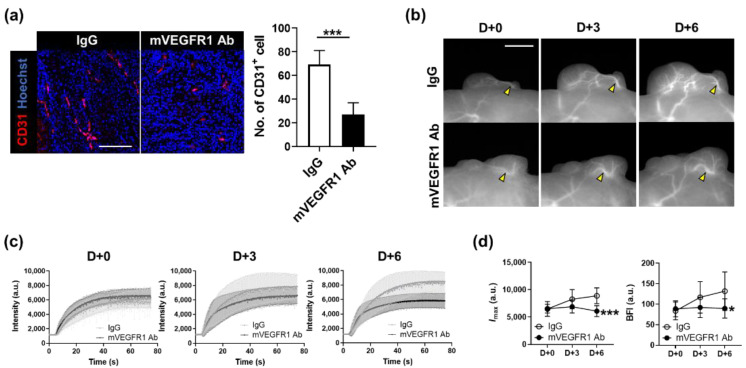
Effect of mVEGFR1 Ab on the thickness of blood vessels and blood flow through main vessels. (**a**) Immunostaining of CD31 (red; left) and the number of CD31-positive cells (right) in A431 tumors treated with IgG or mVEGFR1 Ab. Nuclei are stained in blue with Hoechst 33,342 counterstaining. Data are the mean ± SD for at least three independent fields examined per mouse (*n* ≥ 3 mice per group). Scale bar, 100 μm. ***, *p* < 0.001 by Student’s t-test. (**b**) Longitudinal observation of fluorescence images in tumors treated with IgG or mVEGFR1 Ab twice at D+0 and D+3. Yellow arrowheads indicate blood vessels analyzed. Scale bar, 100 μm. (**c**) ICG dynamics of tumor blood vessels marked by yellow arrowheads in (**b**), from each day. (**d**) Changes in *I*_max_ and BFI in blood vessels by day. Data are the mean ± SDM for *n* ≥ 7 mice per group. *, *p* < 0.05; ***, *p* < 0.001 by two-way ANOVA.

## Data Availability

All data is contained within the article.

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
