# Peer review of "Real-Time Longitudinal Evaluation of Tumor Blood Vessels Using a Compact Preclinical Fluorescence Imaging System"

_biosensors, 2021, doi:10.3390/bios11120471_

Round 1
Reviewer 1 Report
The authors can see comments and suggestions in the attached document.

Reviewer 2 Report
Jeong et al. came out with a novel technique to image microscopic blood vessels in real time, using indocyanine green dye. This technique is tremendously useful for studying blood vessel development. The authors used this technique to evaluate the effect of VEGF inhibition with mVEGFR1 Ab, and observed decreasing thickness and blood flow of the main vessel compared with the control of just IgG.
Overall the paper is well written, background is clearly described, results and conclusions are well presented and supported.
I have only minor suggestions to improve the paper:
- Line 284: change 'micro blood' to 'micro blood vessel'
- Combine Figure 3 and 4 if possible, so readers don't need to jump back and forth to know where the yellow dot on Figure 3 is that links to the results on Figure 4.
- Figure 5B: please quantify the increase in thickness of the blood vessel with IgG and mVEGFR1 Ab
Author Response
We would like to thank the reviewer for the responses on our manuscript. Please find our point-by-point responses highlighted in blue color.
-----------------------------------------------------------------
Line 284: change 'micro blood' to 'micro blood vessel'
We thank the reviewer for pointing out the error. We changed as the reviewer mentioned in the revised manuscript (line 297).
Combine Figure 3 and 4 if possible, so readers don't need to jump back and forth to know where the yellow dot on Figure 3 is that links to the results on Figure 4.
We appreciate the reviewer’s suggestion. For better readability, we have combined Figure 3 and 4 into Figure 4 in the revised manuscript.
Figure 5B: please quantify the increase in thickness of the blood vessel with IgG and mVEGFR1 Ab
We thank the reviewer’s suggestion. The Imax displays the maximum intensity at each pixel dynamics within the ROI. The larger the value of Imax, the greater the amount (volume) of the fluorescence flow, that is, the thickness of the blood vessel. We have quantified the blood vessel thickness as the Imax graph in Figure 5D of the revised manuscript.
Reviewer 3 Report
This manuscript presented result of fluorescence imaging of tumor using a commercial system. The imaging method is not novel and the analysis of the imaging method isn’t sufficient.
What is the imaging depth of this method? Lack of the imaging depth information makes the evaluation of tumor angiogenesis highly questionable. Where were the blood vessels observed located? Were they in the skin layer or tumor?
How do you compare this method with laser speckle imaging which doesn’t require the injection of an exogenous agent?
“When the tumor size reached the designated grade, ICG (2.5 μg/g; Dongindang Pharmaceutical, Siheung-si, Korea) was injected into the respiratory anesthetized mouse”
-- Was the amount of ICG administrated during each imaging session 2.5 μg/g (of body weight)?
“Conflicts of Interest: The authors declare no conflicts of interest.
-- Are you sure about this claim? Two authors are employees of the company that produce the commercial system.
Author Response
We would like to thank the reviewer for the responses on our manuscript. Please find our point-by-point responses highlighted in blue color.
-----------------------------------------------------------------
What is the imaging depth of this method? Lack of the imaging depth information makes the evaluation of tumor angiogenesis highly questionable. Where were the blood vessels observed located? Were they in the skin layer or tumor?
We thank the reviewer for pointing out the critical part. The imaging depth may vary depending on the type of dye. It is reported that fluorescence dyes with the same wavelength as ICG has a penetration depth of about 5 mm in the tissue [1].
For the reviewer’s further question, the focal depth of the optical system used in this study is long enough for the subcutaneous tumor to come into focus at once. Factors that influence the observations of blood vessels are the penetration depth of the excitation light of the fluorescent reagent and the degree of scattering of the emission light. In case of the ICG, the penetration depth of excitation light is about 5 mm and the penetration depth of emission light is longer than that of excitation light, which is sufficient to observe not only subcutaneous tumor but also the tissue at some depth. Hence, it can be said that what we observed was blood vessels located in the tumor.
How do you compare this method with laser speckle imaging which doesn’t require the injection of an exogenous agent?
We thank for the reviewer’s question. Fluorescence imaging with ICG can be taken only when ICG is injected into the tail vein, whereas laser speckle imaging, as mentioned, does not require the injection of an exogenous agent, so that it is relatively convenient to repeat imaging. Despite the disadvantage that an exogenous agent is required, ICG tumor imaging used in our study has the advantage of superior spatial resolution and deeper imaging depth than laser speckle imaging [2-4].
We have included these comparisons of laser speckle imaging and ICG imaging in the Introduction sections of the revised manuscript (line 56).
“When the tumor size reached the designated grade, ICG (2.5 μg/g; Dongindang Pharmaceutical, Siheung-si, Korea) was injected into the respiratory anesthetized mouse”
-- Was the amount of ICG administrated during each imaging session 2.5 μg/g (of body weight)?
Yes, ICG was administered at a dose of 2.5 μg/g based on the mouse body weight for each imaging session.
“Conflicts of Interest: The authors declare no conflicts of interest.
-- Are you sure about this claim? Two authors are employees of the company that produce the commercial system.
Although two of the authors belong to the company that made the equipment, none of the authors surely declared conflicts of interest.
References for the reviewer
- J. Wang, G. Liu, K. C. Leung, R. Loffroy, P. X. Lu, and Y. X. Wang, "Opportunities and Challenges of Fluorescent Carbon Dots in Translational Optical Imaging". Curr Pharm Des 21(37), 5401-5416 (2015).
- J. D. Briers, "Laser Doppler, speckle and related techniques for blood perfusion mapping and imaging". Physiol Meas 22(4), R35-66 (2001).
- S. Eriksson, J. Nilsson, and C. Sturesson, "Non-invasive imaging of microcirculation: a technology review". Med Devices (Auckl) 7, 445-452 (2014).
- V. Rajan, B. Varghese, T. G. van Leeuwen, and W. Steenbergen, "Review of methodological developments in laser Doppler flowmetry". Lasers Med Sci 24(2), 269-283 (2009).
Round 2
Reviewer 1 Report
Dear authors,
please see the attached file.

Reviewer 3 Report
- The information regarding imaging depth should be included in the manuscript
- Move table 1&2 to Supplement. The technical detail of the system isn’t essential information
- Fig. 2, add numbers in the Time axis
Author Response
1. The information regarding imaging depth should be included in the manuscript.
We thank the reviewer’s comment and we have included the information regarding imaging depth as follows in the revised manuscript (line 127): ‘The fluorescence dyes with the same wavelength as ICG have a penetration depth of about 5 mm in the tissue, which is long enough for the subcutaneous tumor to come into focus at once’.
2. Move table 1&2 to Supplement. The technical detail of the system isn’t essential information.
We agree with what the reviewer pointed out. As the reviewer suggested, we have moved Table 1 and Table 2 to the Supplementary file (now Table S2 and Table S3).
3. Fig. 2, add numbers in the Time axis.
We appreciated the reviewer’s suggestion. The graph in Fig. 2 is a schematic one made to explain each parameter, so there are no separate numbers on the time axis. We hope your consideration of this.